# Whole Genome Resequencing from Bulked Populations as a Rapid QTL and Gene Identification Method in Rice

**DOI:** 10.3390/ijms19124000

**Published:** 2018-12-12

**Authors:** Workie Anley Zegeye, Yingxin Zhang, Liyong Cao, Shihua Cheng

**Affiliations:** 1State Key Laboratory of Rice Biology, China National Rice Research Institute, Hangzhou 311400, Zhejiang, China; workieanley@yahoo.com (W.A.Z.); zyxrice@163.com (Y.Z.); 2Department of Plant Sciences, University of Gondar, Gondar, P.O. Box 196, Ethiopia; 3Zhejiang Key Laboratory of Super Rice Research, Hangzhou 311400, China

**Keywords:** bulk segregant analysis, MutMap, QTL-seq, whole genome resequencing

## Abstract

Most Quantitative Trait Loci (QTL) and gene isolation approaches, such as positional- or map-based cloning, are time-consuming and low-throughput methods. Understanding and detecting the genetic material that controls a phenotype is a key means to functionally analyzing genes as well as to enhance crop agronomic traits. In this regard, high-throughput technologies have great prospects for changing the paradigms of DNA marker revealing, genotyping, and for discovering crop genetics and genomic study. Bulk segregant analysis, based on whole genome resequencing approaches, permits the rapid isolation of the genes or QTL responsible for the causative mutation of the phenotypes. MutMap, MutMap Gap, MutMap+, modified MutMap, and QTL-seq methods are among those approaches that have been confirmed to be fruitful gene mapping approaches for crop plants, such as rice, irrespective of whether the characters are determined by polygenes. As a result, in the present study we reviewed the progress made by all these methods to identify QTL or genes in rice.

## 1. Introduction 

Understanding and detecting the genetic material that controls a phenotype is a key means to functionally analyzing genes as well as to enhance crop agronomic traits [1]. Traditional gene identification methods are dominant approaches for probing the heredity of phenotypic variations in agronomic traits [2]. Many quantitative trait loci or genes with significant properties in their crop species have been efficiently used [3,4], although further steps are required to fine map the candidate region and enlarge the physical map through the accumulation of the high marker density and population size, and then by gene identification and confirmation [5,6]. As a result, the positional cloning approach, which uses a large number of markers and populations for gene identification, is usually time consuming, has a low throughput, and involves long and tiring work. 

Currently, due to the advancement of high-throughput techniques, the price of sequencing has rapidly decreased, and the implementation of crop genomics research to boost yield and other agronomic traits has become possible [7]. Therefore, whole gnome resequencing-based gene isolation has greater benefits and is more accurate in isolating a massive number of polymorphic variations and in the localization and refining of the candidate region, in comparison with conventional gene isolation methods [8,9,10,11]. This is because there is no involvement of marker development, phenotyping and genotyping, which are laborious, inefficient, and overpriced techniques for traditional quantitative trait loci or gene mapping. 

Bulked segregant analysis (BSA), which is the fastest and most accurate way to spot molecular markers connected to our trait of interests, while associated with high-throughput tools, such as next-generation sequencing, will enable a quick pathway method for finding candidate gene locations more efficiently and easily [12]. The fast rise of high-throughput techniques has been enhanced through the integration of both the BSA and whole genome resequencing of pooled DNA to detect a candidate gene or QTL location. Through the preliminary development of a segregate population, two DNA pool samples can be made from offspring with complementary phenotypes and then genotyped by genetic markers, which show the genetic variation (polymorphic) among the maternal and paternal [13]. The bulk segregant analysis method is employed for several crop plants to find vital genes [14,15,16]. As it is described in Table 2, with the advent of sequencing technology, BSA-based whole genome resequencing methods intensely accelerate the process of finding responsible genes for the causative mutation of the phenotypes [17]. First, BSA, together with next-generation sequencing, has been applied in angiosperms to spot causative QTL or genes for the development and leaf color of *Arabidopsis* [18]. Next, several strategies have been advanced for ideal crops, such as rice and *Arabidopsis* [19,20,21,22,23,24,25,26,27,28,29,30]. Successful candidate gene detection using this approach has been used for vital phenotypic characteristics in many other crops, such as maize [31], barley [32] soybean [33,34], cucumber [35], tomato [36], and chickpea [37].

New advances in genotype sequencing techniques are reforming varied features of the genetic study. Advancements in these genotype sequencing technologies have empowered the whole genome resequencing of crop plants, which have consequently turned out to be predictable. Whole genome resequencing approaches guarantee a radical influence on crop enhancement in this difficult time of impending food disaster, associated with the alarming global population increase. One application of whole genome resequencing is finding the responsible genes contributing to mutations in the traits of interest. In this regard, a massive number of methods (Table 1 and Table 2) have been applied, such as SHOREmap, X-QTL, Next-generation Mapping, MutMap, QTL-seq and Specific Locus Amplified Sequencing (SLAF)-seq [17,18,19,21,25,38]. Generally, in this review paper, we summarize the advancement of conventional BSA, MutMap, and QTL-seq mapping strategies in crop genetic genomic and genomic study for rapid gene identification.

## 2. Conventional Bulk Segregant Analysis (BSA)

In crop improvement programs, genetic mapping is conducted on the entire population and analyzed via genetic markers as a means of genetic factor array. It comprises a vast range of sample sizes, along with a large number of molecular markers required to guarantee adequate power in statistical analysis. The heredity of most of the characters or traits of interest are complicated because they are controlled by environmental factors, additive gene actions, and epistasis effects. Therefore, the detection of the genetic factors of interest, such as QTL, has been signaling a very important part in the operation of the phenotype characters. As a consequence, QTL mapping by conventional gel-based genetic markers is time consuming, difficult, and costly. Therefore, bulked sampling analysis is an alternative method that significantly decreases the size and price by making the procedure straightforward, because traditional analysis needs to assay whole entities for the target characters composed from the sampled individuals.

Bulk segregant analysis is a chic QTL mapping strategy that permits the synchronized isolation of genes that affect a specific phenotype [12]. This technique was initially used in an F2 population to spot genetic markers associated with particular traits using Random Amplification of Polymorphic DNA (RAPD) and Restriction Fragment Length Polymorphism (RFLP) genetic markers [12,39]. Beginning with a crossed population (segregating population), individuals are genotyped for their targeted characters, and then two DNA bulks from segregants are made by choosing samples from the extreme tails of a phenotypic normal distribution. The method is especially convenient, while there is no adequate simple sequence repeat marker (SSR) accessible in the target region, and due to small variation in crops, such as pigineonpea groundnut, the detection of polymorphic genetic markers is an additional and challenging duty. With the initiation of innovative biotechnology tools, novel sequencing technology and extra bio-assay approaches, the genomic difference between two complementary bulk samples can be examined at the macro- and micro-molecule levels through low-throughput techniques, such as individual markers, microarrays, and whole genome resequencing methods. At the level of DNA, genomic variation can be analyzed by diverse forms of generic marker, microarrays, and next-generation sequencing [40,41] or whole genome resequencing [42,43]. 

Conventional BSA is typically executed by single genetic markers, mainly PCR-based markers and repeat sequence-based markers [12,44]. Recently, array- or chip-based genotyping (e.g., diversity array technology Single Nucleotide Polymorphism (SNP) array, comparative genomic hybridization/CGH array, Targeting Induced Local Lesions in Genomics (TILLING) array) has become predominant, because a huge quantity of genetic markers can be performed within a short period of time. Like conventional BSA approaches, the microarray-based BSA has been carried out by utilizing high-throughput technologies (Table 1). The number of markers analyzed is far better than that of the conventional means, and it meaningfully refines results and increases the effectiveness [8]. Due to the substantial decline of the price of next-generation sequencing (SNP and Indels), third-generation sequencing, such as de novo assembly and targeted sequencing methods at multiple depths, are currently used for bulked sample genotyping or analysis. 

In general, BSA has both some advantages and some limitations. The method is widely used for organisms that have a genome size of less than 1 Gbp, although it has an inadequate usefulness in plants with a higher genome size, because it is expensive for whole genome sequencing and there is a problem associated with assembling big sequencing data. This mainly depends on comparatively simple characters showing a discontinuous variation. However, characters that show continuous variation, which is determined by genes with dissimilar effects, such as entire sample analysis, will not be successful, as the choice of boundaries encompassing all sympathetic genes of a huge amount of gene positions could be impracticable. In contrast to the conventional techniques of genetic analysis, bulked sample analysis needs to recognize individual performing extreme divergent traits in the targeted population. As a result, genetic analysis by bulked sample analysis might fluctuate from the actual value of the phenotypic variance likelihood of odds value and the additive gene influence, taken as an example. Additionally, BSA is perpetually unsuccessful in recognizing epistatic interactions, and it is more insensible to rare phenotyping mistakes. In general, the main drawback of gene or quantitative trait loci by bulk segregant analysis is that the percentage of phenotypic variation elucidated by the genes or QTLs cannot be valued.

## 3. MutMap

In MutMap [21], a wild type is a mutagenesis by mutagens mostly by Ethane methyl Sulfonate (EMS) to produce a mutant population, and a mutant with a trait of interest is backcrossed with the wild type parent, which is used for the mutagenesis (Figure 1). Then after, individual progeny from the segregating population based on our trait of interest, is selected to make the DNA pool. The pooled DNA, which is taken from each individual mutant progeny, are then subjected to sequencing via next-generation sequencing technologies. The sequenced results, which are short reads, need to be aligned with the reference genome, and the alignment product is used to gather the target region of the causative variation in charge of the trait or the characters. The precondition for this approach is that the genomic section covering the genetic variation exists in the parent, which is used as a reference genome. DNA markers are not needed for MutMap analysis. Based on the SNPs linkage, we can easily visualize the SNP index peaks on the Manhattan graph to detect the candidate region. 

The crucial benefit of this method is its capacity to quickly recognize changes distressing quantitative characters in the crop genome. In this technique, scholars sequence only the DNA bulk obtained from individuals, which displays the recessive parents of the segregating population based on high-throughput technologies, such as next-generation sequencing, and then aligns with the assembled reference genome. Essentially, for this method, only a small number of mapping populations (near to 100) is enough for the scoring of the trait or the phenotype and to identify the relevant variation [21]. Hence, the method evades the time-consuming and labor-intensive approach or traditional single DNA marker-based QTL or gene mapping analysis. This means that it overcomes one of the decisive challenges, which is the most time-consuming method for ascertaining molecular markers for the required characters. Moreover, the procedure facilitates the detection of the mutation that causes the elusive quantitative variations of the traits. Altogether, this indicates that the MutMap technique is a more proficient technique for detecting the causative mutations with the measureable effects, compared to the SHORE map and other pooled DNA-based sequencing techniques [18]. 

In rice, many genes that are responsible for causative mutation have been effectively isolated by the MutMap method, with the combination of BSA and whole genome resequencing. Among the genes identified as responsible for the pale green leaf and genes regarding the semi-dwarf phenotype [21] are: *Pii*, the gene responsible for blast resistance [45]; *OsRR22*, responsible for the salinity tolerance [26]; *sm11*, a gene responsible for the small grain size [46]; and the *LUP* gene, which is responsible for the loose upper panicle [47]. 

In general, the MutMap method allows for the rapid identification of the candidate gene as well as the QTL responsible for the phenotype. The newly anticipated next-generation sequencing-based technique has an affordable price for sequencing and is an efficient, accurate, and quick system for linkage analysis to isolate gene harboring locations, because it evades the tiresome genotyping of huge samples and has the ability to detect smaller candidate regions linked to the character than conventional gene or QTL analysis. 

## 4. MutMap Gap

However, the genetic change that is liable to have mutant characters are identified by next-generation sequencing and alignment to the reference genome parental line, the sequenced mutants display basic genetic differences from the reference genome parental line. As a result, identifying genetic changes in the genome regions, which are absent in the reference genome by simple alignment, is impossible. The genomic detoxification of the major varieties is not precisely aligned in the same genome region because of the breed specificity, so mutations in these regions cannot be detected. To overcome these shortcomings [45], a MutMap-Gap combines whole genome resequencing with de novo assembly, as a new technique that encompasses the explanation of a target genomic location harboring a mutation of interest using the recently reported MutMap method, followed by the assembly of the de novo (Figure 2), alignment and detection of SNPs within the candidate region.

Therefore, to identify the genes that are absent from publicly available reference genome sequences (e.g., the Nipponbare reference genome in rice), MutMap-Gap is an alternative gene identification method. For assembly, the sequence read, mapped with the candidate location, and those unmapped were combined and subjected to assembly by DISCOVAR de novo. Lastly, the sequence read of the pooled DNA, obtained from the segregating progeny that displays the target trait, is aligned with the assembled *contigs* and SNP-index, which were calculated by the MutMap pipeline. Using this method, the blast resistance gene, *Pii*, is identified [45]. A flow chart of the procedure, which illustrates the MutMap-Gap analysis, is given in Figure 2.

However, using the MutMap Gap approach, the gap in the candidate region, delimited by the MutMap technique, can be filled using de novo assembly. De novo assembly rebuilt the candidate break by combining unmapped reads with the short reads aligned to the target [45].

However, using the MutMap Gap approach, the gap in the candidate region, delimited by the MutMap technique, can be filled using de novo assembly. De novo assembly rebuilt the candidate break by combining unmapped reads with the short reads aligned to the target [45].

## 5. MutMap+

This technique is based on the cross of mutated trait and the parent used for the mutagenesis. Then, the M1 progeny is advanced into the M2 and M3 segregating progeny for the evaluation of this segregation’s population (Figure 3). Thus, mutant progeny that are not suitable for selfing with the reference genome parental line show the lethality and sterility that could not be used for this technique. Thus, to overcome this kind of bottleneck [24], MutMap+, which is the extended version of MutMap method that is based on crossing with those that reveal the wild type trait and are known to be in the M2 segregating progeny for the wild type and for mutant trait has been developed. Then, the acquired M3 progeny would openly utilize whole genome resequencing activity. Meanwhile, the method escapes the requirement of backcrossing to the wildtype, so it remains appropriate for the identification of genes or QTL that hinder artificial hybridization. Therefore, this technique enlarges the principle of the MutMap method to enhance crop yield and yield-related traits, including rice crop, because of its ability to ascertain causative genetic changes in the primary growth of lethality, infertility—in general, anything that hinders selfing.

Generally, the method has a prolonged application for other organisms, as long as the precondition of backcrossing to the parental line is satisfied. At this juncture, the technique exploited the rice mutant population obtained by EMS mutagenesis to quickly isolate the responsible gene that causes genetic changes in the phenotype or the character using next-generation sequencing and comparison with the Δ (SNP-index) Manhattan plots of pooled DNA found in the M3 segregating population [24]. Remarkably, the substantial benefit of this approach is that it does not require backcrossing with the wildtype parental line, and hence, it offers innovative solutions that are not inconceivable for the unusual MutMap procedure. 

## 6. Modified MutMap 

As described earlier, using the previously mentioned techniques of gene or QTL identification, researchers have isolated mutant genes perfectly, expediently and more significantly than the traditional individual marker-based or positional cloning methods. [21,26,45,48]. Scholars have sequenced only bulked DNA from the F2 segregating progeny that reveal mutant phenotypes based on next-generation sequencing high-throughput technologies using a MutMap approach, then by alignment to the reference genome of the wild type parental line. The population applied to the modified MutMap scheme is comprised of BC1F2 segregating lines, which display a clear seclusion of the mutants and wild type parent. Remarkably, the MutMap technique entails amassing the wild type genome sequence, exactly recycled as the genome reference. As a result, this approach has more false positive effects for various reasons, including striving to define the size of the F2 segregating population presenting recessive traits and determining the average genome coverage depth to sort it out in the wild type and mutant traits [21,49]. To moderate this error, a modified MutMap has been exploited for the effective detection of the WB1 gene responsible for endosperm development in rice [30]. The larger the size of the segregating population displaying the mutant character to be pooled, the higher the genome coverage, the more precise the categorization of the wild type and mutant progeny, and the lower the level of false positives [21], which were a challenge for the MutMap method, but are solved in the modified MutMap method. 

In this method, DNA from both recessive and dominant BC1F2:3 populations were pooled and used for whole genome resequencing (Figure 4). BC1F2:3 populations are more uniform in their genetic makeup, compared to BC1F2 lines, since they reduced the effect of other genetic changes on the trait of interest. As a result, it is simple to discriminate wild and mutant types. Therefore, the modified MutMap approach [30] has exhibited advancement precisely in the high peak in the chromosomal position, compared to that based on Δ (SNP index) [21]. Generally, this technique showed a low error rate, high specificity, and the lowest rate of false positives.

## 7. QTL Seq

In nature, most of the agronomically important crop traits are measured/controlled by polygenes/multiple genes, in which every gene has minor effects on the phenotype, but the commutative effect of each gene contributes to the quantitative trait. Quantitative trait loci mapping is an extremely concrete technique for the genomic segmentation of continuous characters and it offers a preliminary idea for the positional cloning of linked genetic factors or genes, and more importantly, it has a remarkable performance in marker-based selection in crop improvement programs [50]. Nevertheless, the method involves the enlargement and choice of genetic markers for QTL mapping analysis, and it is therefore inefficient and tiring [51] due to the marker designing, identification of polymorphic marker, and genotyping. Conventionally, genes have been isolated by QTL mapping of segregation population progeny, resulting from two parents displaying complementary phenotypes for our attribute of interest. 

Molecular markers that have a capacity to discriminate parental lines are compulsory for QTL mapping. As a result, the two parental lines should be chosen from unrelated genetic backgrounds. The obvious differences between the parents and many QTL determining the traits complicate the identification of the specific gene position. Furthermore, when parents with a genetically similar background are used, the detection of abundant polymorphic markers, associated or linked to our traits of interest, will be very limited. Positional cloning (map-based cloning), by which many rice genes have been identified, is the most commonly used method, but it is a tricky, time-consuming and costly method. The QTL Seq method (Figure 5), which is the new method based on next-generation sequencing, has been exploring rapid QTL or gene isolation [25]. The application of the whole genome resequencing of pooled DNA, obtained from extremely different phenotypes, has been testified in yeast [52,53,54,55,56].

The benefit of ascertaining huge quantities of SNPs by whole genome resequencing is promising for uncovering and enhancing the detection of target chromosomal regions harboring causative mutations, and it is more powerful than conventional single marker linkage analysis methods [9,10,11,57]. The current projected high throughput-based QTL-seq technique is cheap, quick, and accurate, and the efficiency of the gene or QTL identification approach has been confirmed by detecting the causal gene in rice [25]. Furthermore, in this approach, the molecular marker, positioned within the harboring target chromosomal region, can be additionally used to narrow down the region by changing them into more economical Cleaved Amplified polymorphic (CAP) markers for marker-assisted selection breading. Generally, the QTL-seq approach combined with bulk segregant analysis (BSA), followed by whole genome resequencing technologies, is faster and more effective than the past systems applied for QTL detection [25]. As mentioned in the introduction of this paper and Table 1 and Table 2, there are currently many available methods that combine the BSA of mixed DNA to make next-generation sequencing of responsible genes or QTL more efficient and rapid. Among these techniques, many QTL and genes using QTL-seq technique for different rice phenotypes [25,27,29,58,59,60], cucumber [35], tomato [36], chickpea [37], and soybean [33,34] have been effectively isolated. 

## 8. Conclusions

The identification of genetic factors controlling the character is one of the foremost technologies for the characterization of the gene function for the improvement of crop agronomic traits in crops, although conventional analysis strategies, such as single DNA marker-based genotyping, remains a generally low-throughput, long and costly method. Therefore, it should be complemented with whole genome resequencing and the basic DNA sequencing with target enhancement or the reduction in genome difficulty. MutMap, MutMap Gap, MutMap+, modified MutMap, and QTL-seq might hasten the identification of genes or QTL for the trait of interest. However, the phenotype is determined by polygenes. The concepts of the gene or QTL identification tool in all approaches that target the causal genetic variation implement an analogous approach. BSA is used for MutMap; an extreme phenotype is separated into two pools; one pool of samples has a mutant phenotype, and the other consists of samples with a wild type phenotype; while in the case of the QTL seq method, the extreme highest and lowest progeny are selected from the normal distribution, followed by mixing an equal amount of DNA from each pool to build up two extreme pools. Then, each pool is sequenced and assembled, and gene annotation is conducted. The overall objective is to identify whether the polymorphism is SNP or indel, which has only non-reference reads that are responsible for the causative mutation of the phenotypes. In general, further works have to be done in the future for gene or QTL identification in rice especially by using MutMap+ and MutMap Gap methods. 

## Figures and Tables

**Figure 1 ijms-19-04000-f001:**
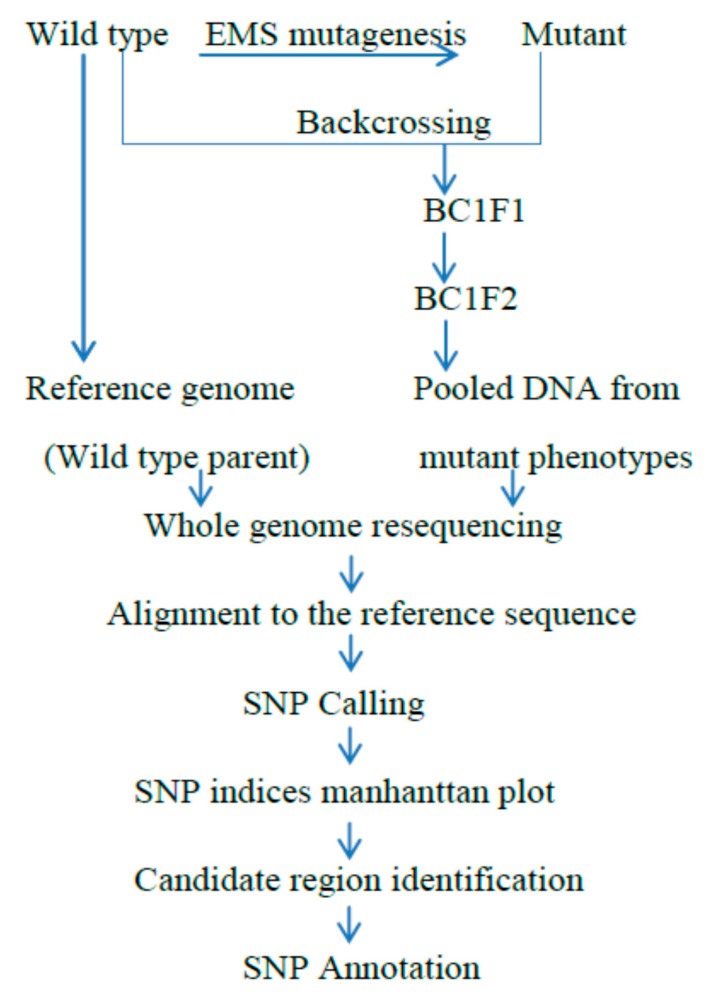
Basic outline of the MutMap method: A wild-type plant, which is used as a reference genome, is treated by the Ethane methyl Sulfonate (EMS) chemical to produce a mutant population. The produced mutant is backcrossed to the wild type of the same plant used for the mutagenesis. Then, BC1F_1_ progeny is self-pollinated to get BC1F_2_ segregating progeny for the mutant and wild-type phenotypes. The backcrossing of the mutant to the wild-type plant confirms the uncovering of the phenotypic changes at the BC1F_2_ progeny between the mutant and wild type. An equal amount of DNA from the BC1F_2_ generation shows that the mutant characters are pooled and evaluated for whole genome resequencing, then by alignment to the genome reference, and finally by determining the Δ SNP index of 1 to detect the candidate genome region of the mutant phenotype [21].

**Figure 2 ijms-19-04000-f002:**
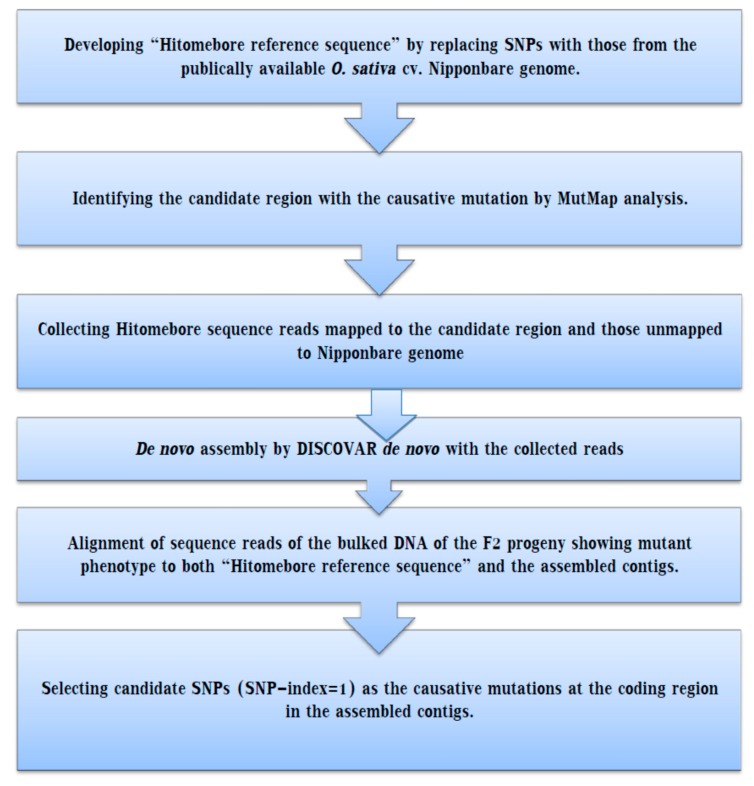
Basic outline of the MutMap gap method: The mutant lines obtained from the EMS-treated wild-type parent are grown to give M1. For MutMap analysis, M1lines are backcrossed with wild type to produce F2 populations that segregate the recessive and dominant characters. DNA samples from this population are pooled in an equal ratio, and then whole genome sequencing is carried out, followed by the assembly of the reference genome. Measuring of SNP index is conducted at each particular SNP position, and an SNP index Manhattan graph is plotted to ease the delimitation of identified genomic regions harboring the candidate gene responsible for the mutation. If the genetic variation is found in the wild type specific chromosomal position, the MutMap method would not isolate the causal mutation alone.

**Figure 3 ijms-19-04000-f003:**
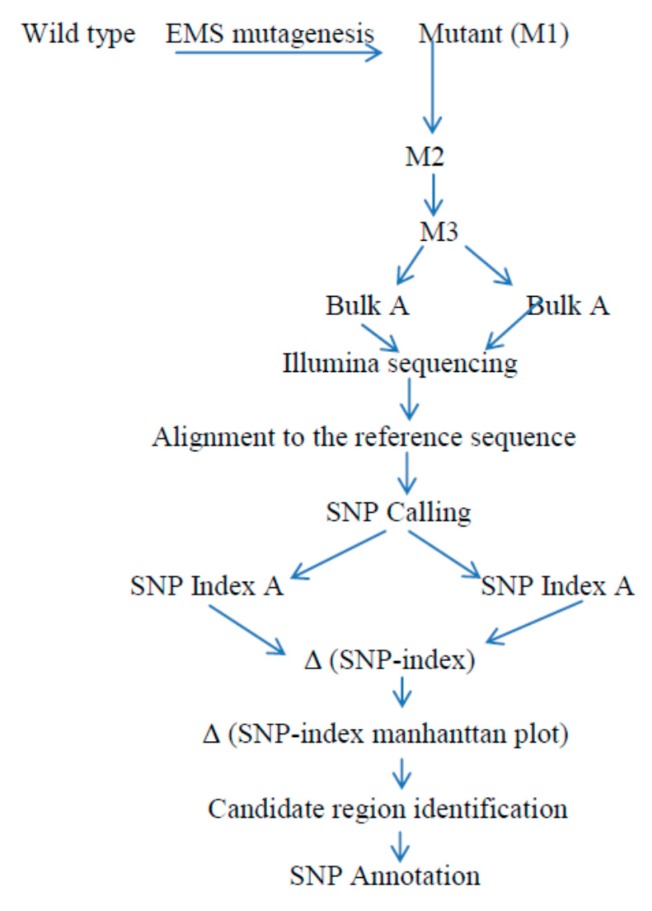
Basic framework of the MutMap+ protocol: following the wild type EMS mutagenesis, seeds are planted to obtain the M1 progeny and M2 generations, which are produced from M1 selfed lines. Again, the heterozygous M2 progenies are self-pollinated to obtain M3 plants. Then, DNA samples from M3 mutants and wild type plants from each sample in an equal ratio are taken to produce pooled DNA, and these bulked DNA are subjected to next-generation sequencing, followed by alignment to the genome reference wild type used for mutagenesis. The intended SNP index at a particular SNP position and Manhattan graph linking SNP index verses chromosome positions are drawn to identify the candidate region that harbors the responsible genes with Δ (SNP index = 1).

**Figure 4 ijms-19-04000-f004:**
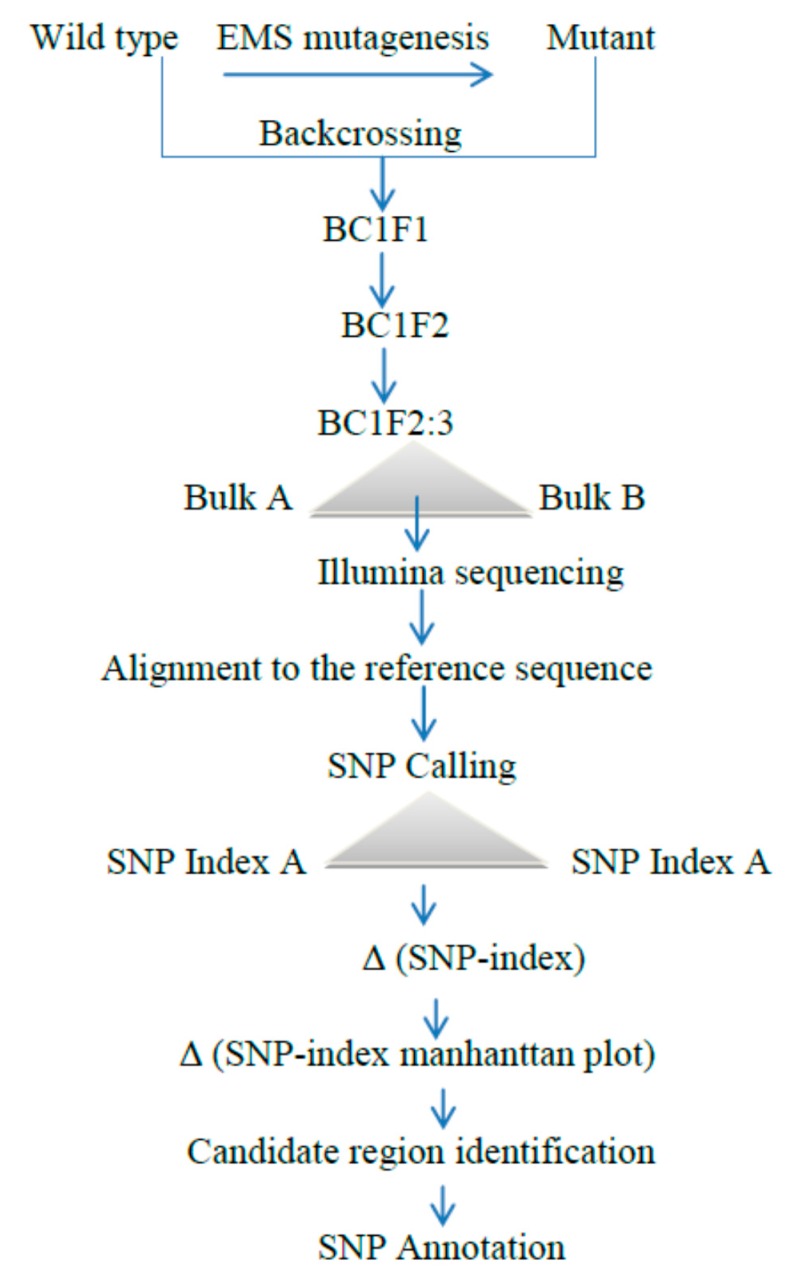
A simplified modified MutMap method scheme: Initially, the mutant was generated from the EMS-treated wild-type variety, then the M2 population was developed from the selfing of these M1 wild types and backcrossed with the mutant to produce BC1F1. The resulting first initial generation of BC1F1 progenies were self-pollinated to obtain BC1F2. From these populations, plants exhibiting the mutant and the wild type were selected to make pooled genomic DNA for whole genome resequencing, followed by alignment and average SNP indices, are estimated for the Manhattan plot. Finally, a candidate region responsible for the phenotype can be obtained by SNP annotation [30].

**Figure 5 ijms-19-04000-f005:**
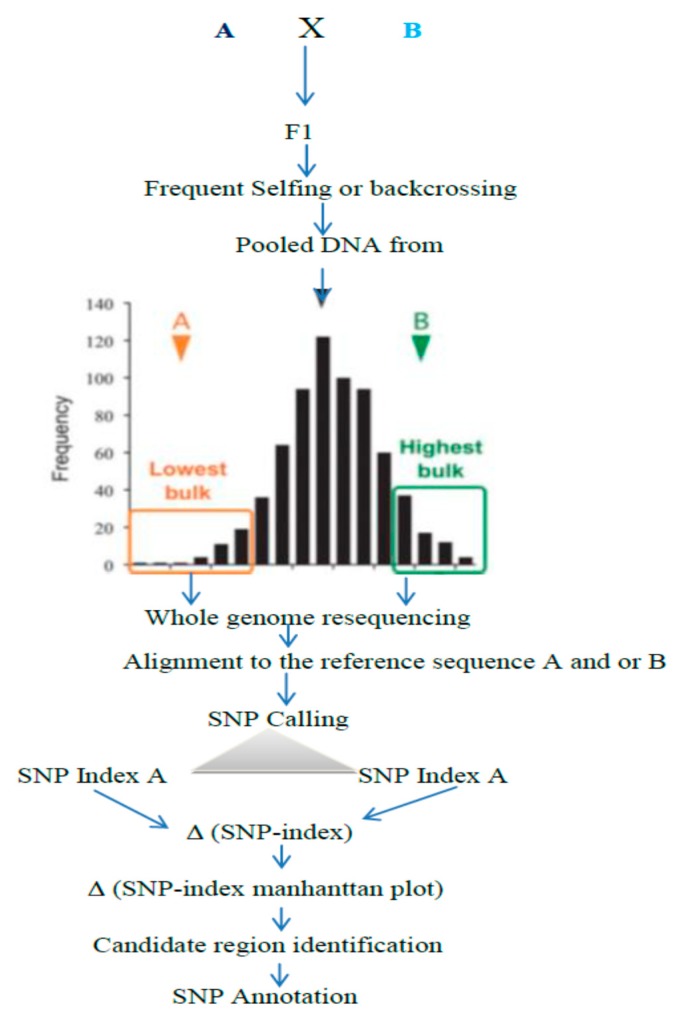
Summarized QTL-seq framework: parents with different genes intercross each other to obtain F1 and the subsequent segregating population. From the segregating population to obtain bulk DNA, samples from the “Highest” and “Lowest” or from the extreme phenotypic values are taken. Then, the pooled DNA are subjected to whole genome resequencing, followed by alignment to the reference sequence to compute the average SNP indices, which are used for the Manhattan plot to identify the candidate region [25].

**Table 1 ijms-19-04000-t001:** Comparison of the conventional BSA, MutMap, MutMap Gap, MutMap+, modified MutMap and QTL-seq methods, based on throughput technology for analytical methods, samples for pool construction, type of population used, presence or absence of unmapped reads, the involvement of SNP-index, Δ(SNP-index) plots and backcrossing.

Mapping Methods	Throughput	Pooled Samples	Population Type	Backcrossing to the Wild Type Parent	Presence or Absence of Unmapped Reads	Use of SNP-Index	Δ (SNP-Index) Plots
Conventional BSA	Low	Extreme tails	All	Yes	Absence	No	No
MutMap	High	Mutant-type phenotype	F2	Yes	Absence	Yes	No
MutMap Gap	High	Mutant-type phenotype	F2	Yes	Presence	Yes	No
MutMap+	High	Mutant and wild-type phenotype	M3	No	Absence	Yes	Yes
Modified MutMap	High	Mutant and wild-type phenotype	F2:3	Yes	Absence	Yes	Yes
QTL-Seq	High	Extreme tails	Segregating population	Yes	Absence	Yes	Yes

**Table 2 ijms-19-04000-t002:** Examples of genes or QTL, identified by using the conventional- or next-generation-based BSA methods in rice.

Sequencing	Traits	Gene or QTL	Population Type	Population Size	Pooled Samples Size	References
DNA-seq	Pale green leaves and semidwarfism	*osCAO1*	F2	-	20, 20	[21]
DNA-seq	Cold tolerance	*qCTSS-1, qCTSS-2b* and *qCTSS-8*	F3	10,800	430, 385	[61]
DNA-seq	Blast disease and seedling vigour	*qPHS3-2*	F2	531	50, 50	[45]
DNA-seq	Early stage lethality	*OsNAP6*	M3	223	40, 40	[24]
DNA-seq	Blast disease	*Pii*	F2	112 and 78	20, 20	[25]
DNA-seq	Salt tolerance	*OsRR22*	F2	67	20, 20	[26]
DNA-seq	Loose Upper Panicle	*LUP*	F2	750	20, 20	[47]
DNA-seq	Small grain size	*smg11*	F2	-	50, 50	[46]
DNA-seq	Endosperm development	*WB1*	F2:3	1000	50, 50	[30]
DNA-seq	Nitrogen Use Efficiency in Rice	*qNUE6*	F2	280	30, 30	[58]
DNA-seq	Cooked grain width and length	*GWi11.1*	landrace accessions	591	-	[62]
DNA-seq	Grain length & Weight	*qTGW5.3*	NIL F2	176	35, 35	[58]
SLAF-Seq	Grain Weight	*qTGW3.2, qTGW3.1* and *qTGW3.3*	RILs	234	30, 30	[50]
DNA-seq	Dwarfness	*asd1*	F2	181	20, 20	[60]
50K SNP Chip	Salt tolerance	*Saltol*	RILs	216	30, 30	[63]
Conventional BSA/individual markers	Male sterility		*pms1*	946	-	[64]
Conventional BSA/individual markers	Small Round Grain	*dep2-3*	F2 and F3	980	8, 8	[65]
Conventional BSA/individual markers	Tiller Suppression	*ts1*	F2	279	10, 10	[66]
Conventional BSA/individual markers	CMS-WA cytoplasm	*Rf*	BC1 & F2	319	6, 15	[67]
Conventional BSA/individual markers	Aroma	*AG8-AR*	F2	125	8, 8	[68]

RILs: Recombinant inbred lines; NIL: near isogenic lines; CMS-WA (Cytoplasmic Male sterility of Wild Abortive type; SLAF Seq: specific locus amplified sequencing, ‘–’: information unavailable.

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
