# Peer review of "Whole Genome Resequencing from Bulked Populations as a Rapid QTL and Gene Identification Method in Rice"

_ijms, 2018, doi:10.3390/ijms19124000_

Round 1

Reviewer 1 Report

The manuscript proposed reviews modern methods for mapping individual genes and QTL responsible controlling agronomically important traits using bulk segregant analysis based whole genome resequencing on the example rice crop. Such a review is undoubtedly very timely taking into account a fast progress in the field and a lot of publications especially concerning rice crop. The manuscript is written on 14 pages, includes Abstract, 8 paragraphs and Reference list. The manuscript is illustrated with 5 figures and two informative tables which contain schemes of MutMap, MutMap Gap, MutMap+, modified MutMap and QTL-seq methods based on throughput technologies, and also compare their main principles and show examples of application in rice.

Unfortunately, the manuscript is abound grammar errors (for example, on the lines 56, 107, 116, 119, 131, 148, 263,267, 274, 285-286, 317 and others) that significantly complicates reading. Moreover, it contains erroneous expressions (for example: “characters or trait of interest are controlled by environmental factor…” – lines 71-72; “our mutant trait” – lines 198, 204”). Therefore the manuscript proposed needs careful editing.

I consider that the article can be published after corrections.

Author Response

Dear reviewer first of all we would like to thank you very much for your constrictive comments and suggestions for the improvement and betterment of the paper. We checked point-by-point and now all the comments and suggestion given by you as well as by other reviewer are incorporated and corrected. More importantly, we submitted to native English speakers for the English grammar errors correction and they did Extensive editing as a result the issue of grammar errors and punctually problem have been solved.

Line.56

Response: New advances in genotype sequencing techniques are reforming varied features of the genetic study.

Line.107

Response: This mainly depends on comparatively simple characters showing a discontinuous variation. However, characters that show continuous variation, which is determined by genes with dissimilar effects, such as entire sample analysis, will not be successful, as the choice of boundaries encompassing all sympathetic genes of a huge amount of gene positions could be impracticable.

Line.116

Response: In general, the main drawback of gene or quantitative trait loci by bulk segrigenat analysis is that the percentage of phenotypic variation elucidated by the genes or QTLs cannot be valued. 

Line.119

Response: In MutMap, a wild type is a mutagenesis by mutagens mostly by EMS to produce a mutant population, and a mutant with a trait of interest is backcrossed with the wild type parent, which is used for the mutagenesis.

Line.131

Response: The crucial benefit of this method is its capacity to quickly recognize changes distressing quantitative characters in the crop genome. In this technique, scholars sequence only the DNA bulk obtained from individuals, which displays the recessive parents of the segregating population based on high-throughput technologies, such as next-generation sequencing, and then aligns with the assembled reference genome.

Line.148

Response: Among the genes identified as responsible for the pale green leaf and genes regarding the semi-dwarf phenotype are: Pii, the gene responsible for blast resistance; OsRR22, that responsible for the salinity tolerance; sm11, a gene responsible for the small grain size; and the LUP gene, which is responsible for the loose upper panicle.

Line.263

Response: In nature, most of agronomically important crop traits are measured /controlled by polygenes/multiple genes, in which every gene has minor effects on the phenotype, but their commutative effect of each gene contributes to the quantitative trait.

Line.267

Response: Quantitative trait loci mapping is an extremely concrete technique for the genomic segmentation of continuous characters and it offers a preliminary idea for the positional cloning of linked genetic factors or genes, and more importantly, it has a remarkable performance in marker-based selection in crop improvement programs.

Line.274

Response: Molecular markers that have a capacity to discriminate parental lines are compulsory for QTL mapping. As a result, the two parental lines should be chosen from unrelated genetic backgrounds.

Line.285-286

Response: The benefit of ascertaining huge quantities of SNPs by whole genome resequencing is promising for uncovering and enhancing the detection of target chromosomal regions harboring causative mutations, and it is more powerful than conventional single marker linkage analysis methods.

Line.317 

Response: The identification of genetic factors controlling the character is one of the foremost technologies for the characterization of the gene function for the improvement of crop agronomic traits in crops, although conventional analysis strategies, such as single DNA marker-based genotyping, remains a generally low-throughput, long and costly method. Therefore, it should be complemented with whole genome resequencing and the basic DNA sequencing with target enhancement or the reduction in genome difficulty.  

      And others are accordingly

Reviewer 2 Report

The manuscript entitled "Whole Genome Resequencing From Bulked Populations as a Rapid QTL and Gene Identification Method in Rice" is a very well described review article. The authors have made a significant effort in compiling all the literature available on rice till date. But still, much recent work is still not cited throughout the manuscript. I would request the authors to also include all the recent articles as a part of this review. Minor changes are required before accepting the article. 

1.    Throughout the manuscript, there are unwanted spaces between two words, punctuations and language need fine revision and also spell check is compulsory throughout the manuscript.

2.    The Abstract needs more clearance about what does actually authors want to say to the readers through this review.

3.    Figure 2, the sentences inside the flowchart is not visible. A higher resolution of the flowchart is required.

4.    The conclusion of the review needs more concrete efforts and what are the future steps must be taken in the Gene identification steps in this cereal.

5.    The English language needs serious revision.

Author Response

Dear reviewer first of all we would like to thank you very much for your constrictive comments and suggestions for the improvement of the paper. We checked point-by-point and now all the comments and suggestion given by you as well as the other reviewer’s comments are incorporated and corrected. More importantly, as per the recommendation of you, we submitted to native English speakers for the English grammar errors correction and they did Extensive editing as a result to issue about grammar and punctually problem.

Point 1. Throughout the manuscript, there are unwanted spaces between two words, punctuations and language need fine revision and also spell check is compulsory throughout the manuscript.

Response 1. We checked thoroughly and we avoided the unwanted spaces between two words and punctuations throughout the manuscript.

Point 2. The Abstract needs more clearance about what does actually authors want to say to the readers through this review.

Response 2. Our objective is to summarize the progress made by all whole genome resequencing based on BSA methods to identify QTL or genes in rice so it is incorporated in the section now.

Point 3. Figure 2, the sentences inside the flowchart is not visible. A higher resolution of the flowchart is required.

Response 3.  Already inserted